

# Technical note: The Weibull distribution as an extreme value alternative for annual maxima

Earl Bardsley[1]

[1] School of Science, University of Waikato, Hamilton 3240, New Zealand

*Correspondence to*: Earl Bardsley (earl.bardsley@waikato.ac.nz)

**Abstract.** The generalized extreme value distribution (GEV) for largest extremes is widely applied to single-site annual maxima sequences for estimating exceedance probabilities for application to design magnitudes under conditions of stationarity. However, the GEV is not the only mode for application of classical extreme value theory to recorded maxima. An alternative approach is to apply specific transformations to the maxima, enabling different but equivalent exceedance

statements. For example, the probability that annual flood maxima will exceed some magnitude $\varepsilon$ is the same as the probability that reciprocals of the maxima will be less than $1/\varepsilon$. The transformed maxima considered here represent sample minima, where the "sample" is the number of transformed independent individual events per year. For sufficiently large sample sizes, this leads to just one of the extreme value distributions for design purposes – the Weibull distribution for minima. This extreme value distribution arises because it is the limit stable expression for describing distributions of large-

sample minima when a lower bound is present.

There is no way of telling whether a good Weibull fit to transformed annual maxima indicates that sample sizes are sufficiently large for the Weibull extreme value approximation to apply. It could happen that a good fit is simply a fortuitous empirical matching to data from transformation selection. However, a similar issue also applies to the GEV which is itself a flexible distribution capable of empirical matching to data.

It is not possible to make a case for the Weibull distribution by application to a range of annual maxima because any number of different transformations might be applied to achieve good Weibull fits. Instead, two simple synthetic examples are used to illustrate how a good fit to annual maxima by the GEV could lead to an incorrect conclusion, in contrast to the Weibull approximation applied to the same examples.

## 1 Introduction

By way of introduction, the Gumbel plot shown in Fig. 1 is comprised of a 100 synthetic data values, taken as representing a sequence of some measure of annual rainfall maxima in arbitrary units. The continuous line passing through the points is a curve of a generalized extreme value distribution (GEV) of largest extremes, with the curve form in this case being of extreme value Type 3 (EV3). This form of GEV has an upper bound parameter and the plotted curve through the data in this



instance gives an upper bound estimate of 4.34 units. This value will change a little with different fitting methods, but the EV3 interpretation would remain that the fit to the maxima indicates the presence of an upper bound.

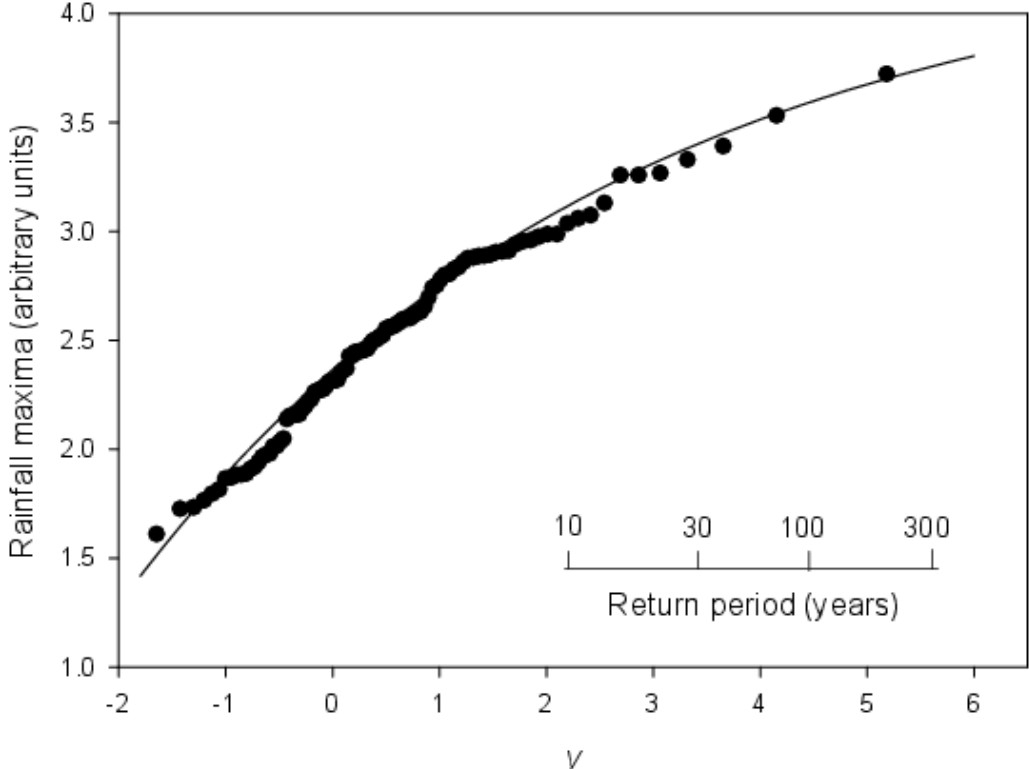

**Figure 1:** Gumbel plot of 100 simulated annual maxima, generated as the largest members of 100 samples of 50 random variables from the standard half-normal distribution. The Gumbel reduced variate is denoted by $y$ and values of $p$ were estimated from the Gringorten plotting expression.

Suppose however that this was a situation of real data and a hydrologist has good independent information that a rainfall upper bound should not be present. The interpretation would then be that the EV3 fit to the data is fortuitous and the necessary conditions for applying the GEV must not be applicable to this data set. For example, it might be that the sample size (number of independent rainfall events per year) is not sufficiently large for the GEV to apply.

The sample size factor is indeed the issue here because the data were simulated at the largest members of 100 samples of size 50, with the samples generated as sets of 50 random variables from the standard half-normal distribution with distribution function:



$$F(x) = \text{erf}(x/\sqrt{2}) \tag{1}$$

where erf denotes the error function. The apparent EV3 form of the maxima in the Gumbel plot of Fig. 1 is an artefact in this
case because it is known that for sufficiently large samples drawn from the normal distribution the sample maxima will
approximate the GEV as a Gumbel distribution (EV1), which plots linearly on a Gumbel plot.

The sample size of 50 is therefore too small to allow convergence of the sample maxima to the EV1 limit distribution. This
is confirmed in Fig. 2 which gives the curve on a Gumbel plot as obtained from the exact distribution of maxima from
samples of size 50 drawn from the standard half-normal distribution. That is, the apparent EV3 curve in Fig. 1 is not derived
from an EV3 distribution with an upper bound, but reflects instead the curve for the unbounded exact distribution of maxima
from samples drawn from the half-normal distribution. The evident EV3 form of the curve is also maintained for normal
distribution maxima for sample sizes larger than 50 (Koutsoyiannis, 2004).

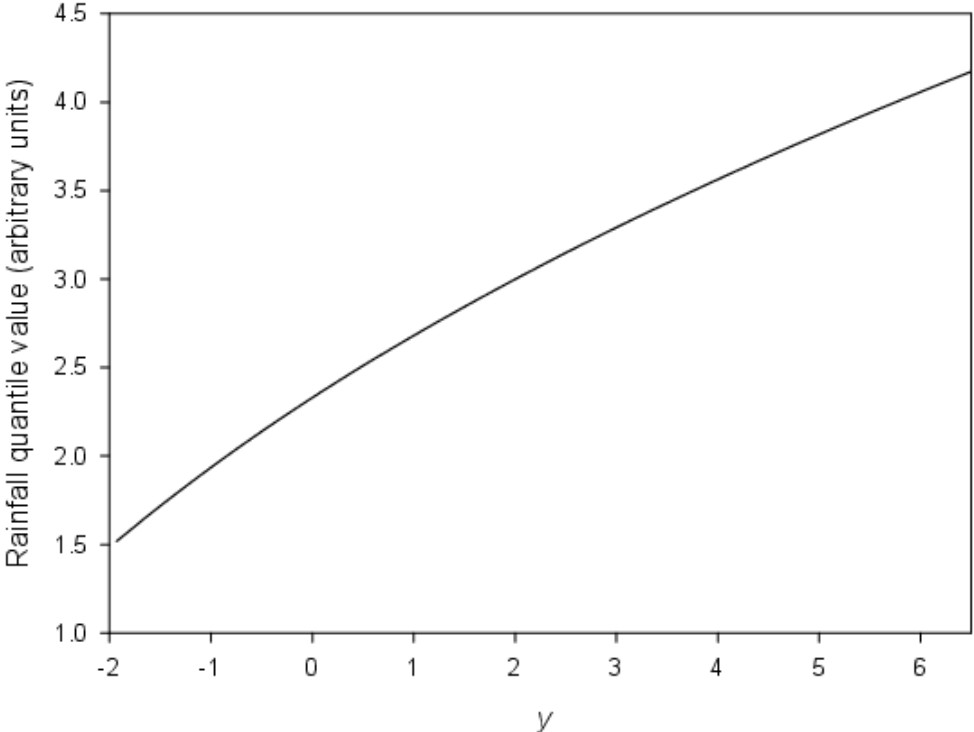

**Figure 2:** Gumbel plot for the exact distribution of maxima of samples of size 50 from the standard half-normal distribution.



In reality the distribution being sampled to provide the maxima is always unknown but a hydrological argument such as an EV3 upper bound being unacceptable could cause rejection of the GEV model, even if it is good fit to annual maxima data. Faced with situations of this kind, a possible alternative extreme value model is outlined here. The method is not in any way an advance of extreme value theory, but uses classical theory applied to a class of transformations of recorded maxima.

## 2 Transformation method

Define $X_1$. $X_2$, ..$X_N$ to be a sequence of recorded annual maxima, assumed to be independent random variables, and define the transformation:

$Y_i = g(X_i)$                                                                                        (2)

such that such that the $Y_i$ values are non-negative and in reverse rank order to the $X_i$. The $Y_i$ are therefore subject to a non-negative lower bound. The transformation can be expressed more formally as $g(x)$ being a continuous positive decreasing function of $x$. Because the $X_i$ are independent random variables then this must hold for the $Y_i$ also. The transformation would
include, for example, $Y_i = 1/X_i$, but not $Y_i = - X_i$.

The transformation has the effect of converting analyses of annual maxima to alternative analyses of annual minima. For example, the probability that annual flood maxima will exceed some magnitude ε is the same as the probability that reciprocals of the annual maxima will be less than 1/ε.


If distributions of annual maxima were known exactly then transformations would serve no purpose. However, the reason for applying extreme value models is to obtain approximate distributions for extremes because the exact distributions are never known. The potential of the transformation approach is that a different extreme value model is applied which sometimes may give a better approximation. Specifically, under general conditions the presence of the lower bound to the $Y$ values means
that the Weibull distribution is the appropriate large-sample approximation to the distribution of $Y$. See, for example, Gottschalk et al. (2013) for discussion of the Weibull distribution as a suitable extreme value distribution in the context of annual minimum discharges.

In summary, the proposed method is to transform the annual maxima via some defined $g(x)$ in Eq. (2) and then apply the
Weibull distribution to the transformed values. Design magnitudes and exceedance probabilities can then be obtained if there is a satisfactory fit of the Weibull distribution. Extrapolating beyond the data range requires assuming that the sample size is sufficiently large for the Weibull distribution to hold as the extreme value limit distribution. The same type of assumption also applies when extrapolating beyond the largest data value if the GEV is used to fit annual maxima.



The transformation approach can never be "proved" as being better or worse that the GEV by checking against many different sets of recorded annual maxima, because it is likely there will always be some transformation that can be found which yields a good Weibull fit. Instead, two text-book distribution examples are given in the next section to demonstrate

that it is possible for simple transformations to give an improved extreme value result compared to the GEV applied to the untransformed maxima.

## 3 Examples

The half-normal example is considered first. The $X_i$ data plotted in Fig. 1 were transformed as $Y_i = \exp(-X_i^{\,3/2})$, where $0 \leq Y_i \leq 1$. We now seek to check that the $Y_i$'s can be approximated by a 2-parameter Weibull distribution. This is obviously

an approximation in this case because the Weibull distribution has no upper bound. However, it is the smaller values of $Y_i$ which are of most interest as they correspond to the largest magnitudes of the original data.

A graphical test for a 2-parameter Weibull distribution is to plot the log of the magnitudes against $z$, where $z = \ln[-\ln(1-p)]$ and $p$ is the probability of a smaller $Y$ value as estimated by a plotting position expression. Application to the

$Y_i$ values gives the approximately linear data plot of Fig. 3. That is, the data are now approximated by an extreme value model which does not incorporate an upper bound with respect to the original data. The presence of any upper bound would have been evident by the $Y_i$ values being better described by a 3-parameter Weibull distribution with some positive value of the location parameter. That parameter value would be first subtracted from the $Y_i$ values in order to achieve a linear plot of the log magnitudes against $z$.


The exact distribution of sample minima is known in this case, so the degree to which this exact distribution is approximated by a 2-parameter Weibull distribution can be seen graphically using a similar plot. This plot is created using many quantile values of the exact transformed distribution instead of $Y_i$ values and using exact $p$ values for calculating $z$ instead of plotting position estimates. As can be seen from the near-linear nature of the plot in Fig. 4, the exact distribution function of

transformed values is approximated by the Weibull distribution function over much of the distribution range, which explains the linear nature of the data plot in Fig. 3.




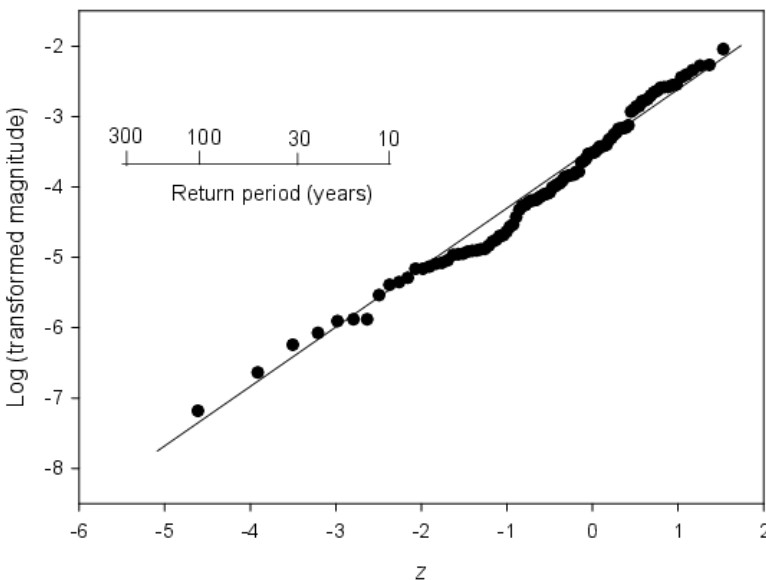

**Figure 3:** Weibull plot of the simulated maxima of Fig. 1 after transforming as $Y = \exp(-X^{3/2})$. Both time and magnitude axes are reversed. The Weibull reduced variate is $z = \ln[-\ln(1-p)]$, where the probability $p$ of a smaller $Y$ value was estimated from the Weibull plotting position expression.

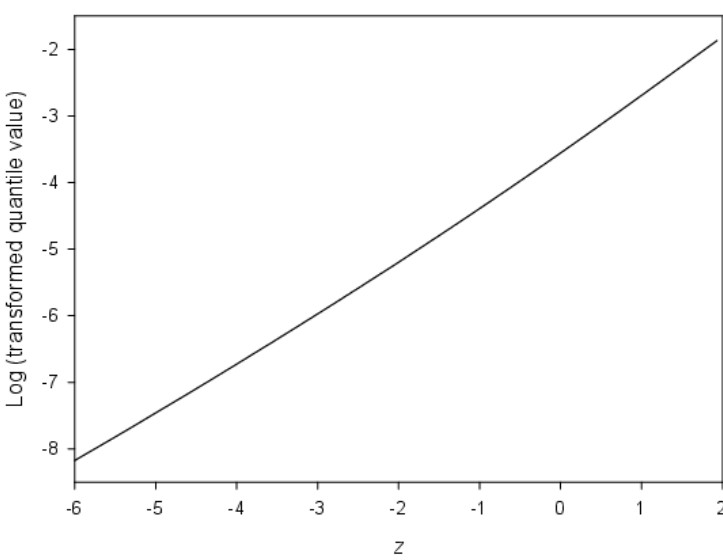

**Figure 4:** Weibull plot for the exact distribution of minima of samples of 50 from the transformed standard half-normal distribution with the change of variable $Y = \exp(-X^{3/2})$.



The second example is brief consideration of the case where the sampled parent distribution of individual events itself follows a specific Weibull distribution with distribution function $F(x) = 1 - \exp(-x^{0.5})$. This distribution was used by Koutsoyiannis (2004) to illustrate the slow convergence of its sample maxima to the limit Gumbel distribution which holds

in this case. In that study, an apparent (incorrect) GEV form of extreme value Type 2 (EV2) was evident even when sample size exceeded $N = 1000$. It was suggested that the EV2 distribution could be employed in an empirical sense for extreme value analysis of annual maxima in such situations, because the limit Gumbel distribution could not be achieved in practice without having an impossibly large number of independent events per year.

As an alternative, applying the transformation $Y = \exp(-X^{0.5})$ to the sample maxima from this Weibull distribution corresponds to $Y$ being distributed as the smallest value from samples drawn from the uniform distribution defined over 0,1. In this case, as $N$ increases the distribution of minima of samples from the uniform distribution approaches the limit Weibull distribution for minima much faster than the original sample maxima approach the limit Gumbel distribution for maxima. Even $N = 10$ is sufficiently large for practical application (Fig. 5), if it actually happened that the parent distribution was the

specific Weibull distribution considered by Koutsoyiannis (2004).

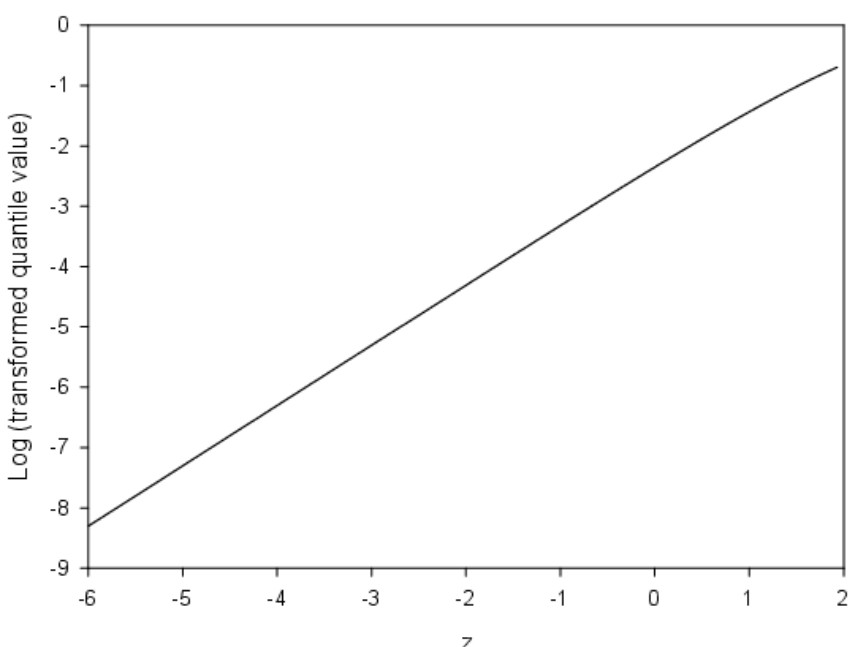

**Figure 5:** Weibull plot for the exact distribution of minima of samples of 10 drawn from the transformed Weibull distribution $F(x) = 1 - \exp(-x^{0.5})$ and with change of variable $Y = \exp(-X^{0.5})$.



## 4 Discussion and conclusion

The two example transformations were selected in the knowledge of the respective true distributions of individual events. However, if distributions were known in this way then transformations would not be needed because exceedance
probabilities could be calculated exactly. In practice, distributions of individual events are inevitably poorly defined by data in the large-magnitude zone of interest. There can therefore be no optimal transformation to recommend here. However, what can be done is to use a flexible transformation like $Y = \exp[-(X/a)^b]$, which in effect means adding two parameters to improve distribution flexibility. If positive constants $a$ and $b$ can be found which yield a good Weibull fit to a set of $Y$ values then the hope is that this reflects convergence to the limit Weibull distribution sufficient to enable some degree of
extrapolation beyond the data range.

It is in the nature of science that if theory gives rise to a flexible model then matching to data cannot be interpreted as confirmation of the model. This is the case with transformations of maxima because it is not possible to exclude fortuitous matching. Similar considerations also apply to the GEV, which is itself a flexible distribution capable of empirical data
matching without the true limit GEV forms necessarily being achieved. In this regard, both the half-normal and Weibull examples considered here will indicate incorrect GEV forms on Gumbel plots unless the sample size is impossibly large in the context of annual maxima.

Conditional on a given transformation, there is no theoretical reason why the distribution of $Y$ should generally converge to
limit Weibull distribution at a slower rate than the distribution of $X$ converges to the GEV. As with the GEV, the transformation approach still depends of classical extreme value theory holding. This may not always happen and it is possible to find mathematical situations where limit extreme value distributions cannot apply. For example, Kotz and Nadarajah (2000, p. 55) cite Green (1976) as noting that maxima of random variables need not always approach stable limit distributions with increasing $N$.


Transforming annual maxima via Eq. (2) to achieve Weibull distribution approximations is probably best viewed as a semi-empirical alternative to the GEV being applied directly to annual maxima. There is no suggestion that transformations to minima should necessarily be used in preference to the GEV distribution of largest extremes. The transformation approach is likely to be of most value when the GEV seems unsuitable on the grounds of either hydrology or fitting to data. Finally, it is
reiterated that the transformation method is simply an alternative application of classical extreme value methods and is not a contribution to theory which builds on recent work.



**Acknowledgement**

This paper contains elements on an earlier paper submitted to HESSD which did not go to final publication but generated helpful reviewer comments.

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
