# Peer review of "Technical note: The Weibull distribution as an extreme value alternative for annual maxima"

_Hydrology and Earth System Sciences, 2018_

## Referee Comment (RC1) · Anonymous Referee #1 · 23 May 2018

If I understand the paper correctly, the author focuses on a model-choice problem arising when estimating an extreme value distribution from not enough data. What might happen is that a distribution is chosen that has, e.g., a finite endpoint (EV3 in the paper) while an infinite endpoint (EV1) would be more appropriate given either knowledge of the true underlying distribution or some plausible knowledge about the nature of the observation.

To circumvent such a model risk, the author proposes to transform the data x by a suitable function g and consider y=g(x) instead. The transformation could be y=1/x for example. The new data would have a positive support and would be bounded by zero. The Weibull distribution in the sense of extremes of minima would be a (sole) candidate for fitting the transformed data for both cases, bounded and unbounded original data.
The author provides two examples.

Of course, the author reduces the model risk (EV1 vs EV3) now. But the author introduces new parameters due to the transformation g. Additionally, largely deviating extreme values (e.g., positive observation with 80, 100, 120 beeing the largest observations) will be transformed to points lying very near now (1/80, 1/100, 1/120) while the body of the data is very prominent after the transformation and highly drives the estimation results. So I wonder how this both effects the estimation accuracy and the implied accuracy of derived quantiles etc. The provided examples do not provide insight into this (and are based on a known g).

So I would like to see a thorough investigation of the applicability and precision of the idea compared to the classical approach; i.e., theoretical results and/or broad simulation study how good can we estimate/extrapolate the distribution of large values of x based on the fitting of Weibull to y=g(x). Note that model risk would also be reduced if one does not allow to fit an EV3 distribution in cases where an infinite endpoint is plausible.

HESSD

---

## Author Comment (AC1) · 12 Jun 2018

My thanks to Referee #1 for comments

The transformation approach is not necessarily better or worse at achieving convergence to a limit extreme value distribution, so it is not advocated as an improved means of reducing risk. Depending on the situation, application of the Weibull distribution to transformed annual maxima could result in greater or lesser risk, or risk largely unchanged. The main point is rather that there is as much extreme value theory justification in applying the Weibull distribution to transformed maxima as there is for applying the generalized extreme value distribution (GEV) directly to annual maxima. Of course, if there are many data-estimated parameters involved in the transformation then there is an increased chance that a Weibull fit to the transformed maxima is just fortuitous data matching and not extreme value convergence. However, fortuitous data fits can apply for the GEV as well.

Estimation via transformed data is not unusual in statistical applications. For example, for estimating a quantile of a lognormal distribution from a lognormal data set, the logs of the data values might be fitted by a normal distribution and the lognormal quantile value estimated as a normal quantile value.

There is certainly a need for further investigations – with respect in particular to seeking if applicable general transformations can be found which lead to relatively rapid convergence to the limit Weibull distribution for a range of different positive-valued text-book univariate distributions. The hope would then be that such transformations yielding Weibull fits to transformed annual maxima are in fact indicative of convergence to the Weibull limit distribution of minima and not simply data matching. However, reporting a full investigation of this type is beyond the scope of a HESS Technical note. Hopefully the present brief communication may encourage others reporting some further work along these lines.

---

## Referee Comment (RC2) · Anonymous Referee #2 · 26 Jun 2018

This paper deals with estimation uncertainty issues arising in Extreme Value Theory (EVT) applications due to finite sample sizes. It showcases how an EV3 fit could be obtained from small sample sizes even for extremes from distributions belonging to the domain of attraction of the Gumbel, presenting as an example the half-normal distribution. The EV3 fit implies the presence of an upper bound for extremes which is generally to be avoided for most geophysical variables. The author proposes the use of a known transformation to obtain sample minima instead, which then could be modeled by the Weibull distribution, which also has EVT justification, and no upper bound. Therefore, this alternative modelling procedure is proposed as a potential remedy, with the author claiming it to have as much EVT support as the straightforward GEV fitting.

The paper emphasizes the important issue of estimation uncertainty and shows how

empirical results could be easily misinterpreted. It also succeeds in conveying the message that data matching does not equal confirmation of theory. However, I find that there is no hydrological context in the paper. There is an attempt to give some hydrological insights by simulating "rainfall extremes" from a half-normal distribution, but this is somewhat unfounded. It is unlikely that the half-normal would be a plausible candidate distribution to model rainfall quantities and I think it is misleading to name outcomes of this arbitrary simulation as "rainfall maxima" in Figure 1.

In the author's words: "Suppose however that this was a situation of real data and a hydrologist...". But obviously this mere hypothesis cannot serve as the motivation for writing a hydrological paper and the paper itself presents no evidence that this could be a real situation. This is not trivial; a paper in order to be within the scope of HESS, should present some contribution to a real hydrological problem. Has the author found this behavior (EV3 fit) in many samples of hydrological variables? If yes, this is indeed of interest and should be presented in the paper and the benefit of the presented methodology should also be demonstrated in this respect. Because usually, Gumbel plots (Figure 1) of hydrological variables show the opposite behavior (towards heavy tails).

If this is not a common hydrological problem and this work is intended as a mere statistical exercise, then I think it is not suited for the audience of HESS and it should be addressed to a more statistical journal, where it would be potentially better appreciated.

---

## Author Comment (AC2) · 30 Jun 2018

My thanks to Referee #2 for comment and constructive suggestions.

Apparent EV3 forms are not necessarily so rare in hydrology and might arise, for example, for annual stage height maxima when river floods extend over a wide flood plain. This is presumably the origin of the EV3 form in the plot of stage height maxima shown in Fig. 3b of Jenkinson (1955).

As an example of an apparent EV3 form for discharge maxima, Fig. 1 shows a Gumbel plot of annual flow maxima recorded in the upper Whanganui River, New Zealand. The fitted EV3 curve gives a discharge upper bound estimate of 73 $m^3s^{-1}$. The maxima $X$ were transformed as $Y = \exp(-\exp(X/a))$, where $a$ is a scale parameter – in this case the mean of the maxima. The transformation results in the distribution of $Y$ being approximated by a 2-parameter Weibull distribution, as indicated by the linearity of the corresponding Weibull plot (Fig. 2).

[Figure]

Figure 1. Gumbel plot of annual flow maxima for the Whanganui River at Te Porere, New Zealand (1966-2017). $y$ is the Gumbel variate.

[Figure]

Figure 2. Weibull plot of the transformed annual flow maxima of Fig. 1. $z$ is the Weibull variate (corresponding to - y).

As described in the paper with reference to the half-normal distribution, an alternative extreme value interpretation for the Whanganui maxima is that the sample size (number of independent flood events per year) is not large enough for the distribution of annual maxima to be approximated by the true unbounded extreme value limit distribution. However, the same sample size is sufficiently large for the distribution of the transformed maxima $Y$ to be approximated by a 2-parameter Weibull distribution of smallest extremes, which is consistent with no upper bound for the original data.  That is, the apparent EV3 form of the annual maxima distribution is seen as an artefact caused by the annual maxima distribution being subasymptotic with respect to the true limit extreme value distribution.  This does not represent proof of the absence of an upper bound for the data. However, the point is made that an equally valid alternative extreme value analysis does not require an upper bound parameter.

There is no suggestion that the double exponential transformation used here has general applicability for apparent Type 3 forms of annual maxima, but it is an obvious starting point. As noted in the paper, different data sets may require different transformations to achieve a Weibull approximation.

There is actually no "author claim" in the paper about applicability of extreme value theory. Classical asymptotic extreme value theory was developed with respect to maxima or minima of large samples of independent random variables drawn from the same distribution. Whether a sampled distribution is derived as a transformation of some other distribution is of no consequence, unless the transformation was deliberately contrived in such a way as to make the asymptotic theory inapplicable. Thus, the paper makes no claim to new theory and its value is solely with respect to its potential for practical hydrological application. My thanks for the requested hydrological example, which should give some encouragement in this regard.

The suggestion that the paper is concerned with small samples is not quite correct. For example, reference is made to a distribution mentioned by Koutsoyiannis (2004), where even a sample size of 1000 would result in a false EV2 form for maxima in a Gumbel plot.

Yes – a "rainfall" label for the half-normal distribution example was unfortunate. It would probably be best to leave it undefined as something along the lines of "Hydrological variable".

References

Jenkinson, A.F.: The frequency distribution of the annual maximum (or minimum) values of meteorological elements. Quarterly Journal of the Royal Meteorological Society, 81, 158–171, 1955.

Koutsoyiannis, D.: Statistics of extremes and estimation of extreme rainfall: I. Theoretical investigation. Hydrol. Sci. J. 49, 575–590, 2004.

---

## Author Comment (AC3) · 8 Jul 2018

My thanks to both reviewers.

By way of summary, classical extreme value theory is concerned with limit distributions of maxima (or minima) of large samples, viewed as samples comprising random variables from some probability distribution. For sufficiently large samples, the GEV distribution of largest extremes applies as the extreme value limit distribution for sample maxima. However, in practical application to annual maxima it can never be known whether the sample (number of independent events per year) is sufficiently large for the GEV approximation to apply. Therefore, a good fit of the GEV to annual maxima might indicate that the sample size is in fact sufficiently large. However, the good fit may also just arise because the GEV is a flexible distribution capable of matching to data.

How "large" a sample is needed depends on the distribution being sampled (unknown in practice). For example, both the normal and exponential distributions are in the domain of attraction of the Gumbel distribution, but much smaller exponential distribution samples are required to achieve the Gumbel limit for sample maxima.

This suggests using data transformation with the possibility of creating a faster convergence to the limit extreme value distribution concerned. The paper defines a class of transformations such that for sufficiently large sample size the annual maxima are approximated as random variables from the Weibull distribution of smallest extremes (provided a limit distribution exists). This applies regardless of whether the original data are in the domain of attraction of Type 1, Type 2, or Type 3 extreme value distributions. As with direct application using the GEV, there is still no certainty as to whether the Weibull limit has in fact been achieved, or simply that the transformation used is sufficiently flexible to achieve a match of the transformed maxima to the Weibull distribution.

There is no new theory involved. The transformation simply converts the extreme value situation of annual maxima to an alternative one of (transformed) annual minima. This excludes the trivial transformation of reversal of sign, which would have no influence on convergence.

Both referees raise the issue of hydrological relevance. In practice, the annual maxima transformation to Weibull is most likely to find hydrological application to apparent EV3-fitted annual maxima, supposedly indicating the presence of an upper bound to the variable concerned. Such data are not so common, but do arise from time to time in hydrology. Provided a transformation can be found that transforms the maxima to be well-approximated by a 2-parameter Weibull distribution of smallest extremes, exceedance probabilities can be obtained without the EV3 necessity of specifying a numerical value for an upper bound parameter. For example, in the response to Referee #2, the EV3 upper bound value of 73 $m^3s^{-1}$ has a small but non-zero exceedance probability with the Weibull model.

Earl Bardsley